# Biomechanical Effects of Different Sitting Postures and Physiologic Movements on the Lumbar Spine: A Finite Element Study

**DOI:** 10.3390/bioengineering10091051

**Published:** 2023-09-07

**Authors:** Mingoo Cho, Jun-Sang Han, Sungwook Kang, Chang-Hwan Ahn, Dong-Hee Kim, Chul-Hyun Kim, Kyoung-Tae Kim, Ae-Ryoung Kim, Jong-Moon Hwang

**Affiliations:** 1Precision Mechanical Process and Control R&D Group, Korea Institute of Industrial Technology, Jinju-si 52845, Republic of Korea; cmg0142@kitech.re.kr (M.C.); swkang@kitech.re.kr (S.K.); 2Department of Rehabilitation Medicine, Kyungpook National University Hospital, Daegu 41944, Republic of Korea; goodream123@gmail.com (J.-S.H.); ach3011@naver.com (C.-H.A.); chgim@knu.ac.kr (C.-H.K.); 3Department of Orthopaedic Surgery, Gyeongsang National University, College of Medicine, Jinju-si 52727, Republic of Korea; dhkim8311@gnu.ac.kr; 4Department of Rehabilitation Medicine, School of Medicine, Kyungpook National University, Daegu 41944, Republic of Korea; 5Department of Neurosurgery, Kyungpook National University Hospital, Daegu 41944, Republic of Korea; nskimkt7@gmail.com; 6Department of Neurosurgery, School of Medicine, Kyungpook National University, Daegu 41944, Republic of Korea

**Keywords:** degenerative disc disease, posture, lordosis

## Abstract

This study used the finite element method(FEM) to investigate how pressure on the lumbar spine changes during dynamic movements in different postures: standing, erect sitting on a chair, slumped sitting on a chair, and sitting on the floor. Three load modes (flexion, lateral bending, and axial rotation) were applied to the FEM, simulating movements of the lumbar spine. Results showed no significant difference in pressure distribution on the annulus fiber and nucleus pulposus, representing intradiscal pressure, as well as on the cortical bone during movements between standing and erect sitting postures. However, both slumped sitting on a chair and sitting on the floor postures significantly increased pressure on the nucleus pulposus, annulus fibrosus, and cortical bone in all three movements when compared to standing or erect sitting on a chair. Notably, sitting on the floor resulted in even higher pressure on the nucleus pulposus and annulus fibers compared to slumped sitting on a chair. The decreased lumbar lordosis while sitting on the floor led to the highest increase in pressure on the annulus fiber and nucleus pulposus in the lumbar spine. In conclusion, maintaining an erect sitting position with increased lumbar lordosis during seated activities can effectively reduce intradiscal pressure and cortical bone stress associated with degenerative disc diseases and spinal deformities.

## 1. Introduction

The lumbar spine comprises three functional elements: the vertebral body, the posterior element, and the pedicle connecting the two. The posterior element is formed by two plates composed of two transverse processes, four condyles, and one spinous process [1]. The internal structure of the vertebral body is ideally designed to withstand high compressive forces acting on the lumbar spine [2]. 

Spinal alignment is critical for maintaining the load on the spine and pelvis. The relationship between the pelvis and spine is generally expressed by three angles: pelvic inclination, pelvic incidence angle, and pelvic incidence minus lumbar lordosis. The average pelvic incidence angle is generally 50° to 55° but varies from 30° to 80°. The pelvic angle of incidence plays an important role in lumbar lordosis [3,4]. 

Lumbar lordosis is a unique structural feature of the normal human spine. Phylogenetically, it is considered a structural adaptation to upright walking and places the torso’s center of mass over the hips, improving the ability to withstand gravity [5,6]. In standing and sitting, movement and posture require changes in the inclination of the lumbar spine and sacrum [7]. When the alignment of the spine changes, the distribution of pressure applied to each spinal segment shifts, causing structural changes in the vertebral body and disc, which can result in disc herniation and spondylolisthesis [8,9,10].

Calculations of intradiscal pressure and lumbar spinal load are widely used to prevent spinal disorders by forming the basis of clinical advice to promote correct sitting posture [11]. Many studies have aimed to measure the pressure distribution within the lumbar spine across various postures. These studies typically assessed the lumbar spine pressure in a static and immobile state within each position [12,13,14,15,16]. However, individuals commonly spend substantial periods standing or sitting, engaging in movements such as flexion, lateral bending, and axial rotation. Hence, comprehending the stress on the lumbar spine during diverse movements within each position, as opposed to only in static postures, is crucial for understanding actual lumbar spine stress. 

It is estimated that adults in the United States spend over eight hours per day seated, illustrating that a significant portion of contemporary individuals conduct their daily activities while seated—working, eating, and engaging in leisure [17]. Furthermore, adopting a cross-legged sitting position on the floor is a prevalent practice in Asian cultures. Lumbar degenerative kyphosis (LDK) is often observed among elderly women in rural Asian regions who primarily adopt the sitting on the floor posture [18,19]. Notably, it is understood that the sitting on the floor posture is linked to LDK [18,20]. 

Therefore, this study investigates the pressure exerted on spinal structures during three movements (flexion, lateral bending, and axial rotation) in a total of four postures (standing, sitting upright in a chair, slouching in a chair, and sitting on the floor), including the addition of sitting on the floor. The study utilizes finite element model (FEM) analysis, and its primary objective is to determine the optimal posture for minimizing stress on the spinal structures during movement.

## 2. Materials and Methods

### 2.1. Development of the FEM

A 3D finite element model (FEM) centered on the lumbar spine was constructed for this study [21,22]. The finite element model was specifically created to represent men in their mid-30s, with an average height of 175 cm. Digital data on the human body were acquired from the Korea Institute of Science and Technology Information (KISTI) and used under an agreement. No Institutional Review Board (IRB) approval was required in this case.

KISTI has been providing Korean human information, including CT scans, MRIs, serially sectioned images, segmented images, 3D images, and X-ray images since the year 2000. A variety of Korean human datasets have been prepared and made available to users both domestically and internationally. The size of the vertebral body finite element model used in this study was compared with experimental data in the existing literature to prove its appropriateness [23].

The vertebral body for each posture was modeled using X-ray images from a previous work and a previously created FEM [24,25,26]. Data on lumbar lordosis and pelvic parameters important for spine alignment in each posture are based on a previous study [24,26]. In particular, the size of the constructed lumbar spine model was valid within the error range when compared with the actual measurement results of a previous study [27]. In this study, we analyzed the stress applied to the lumbar region according to posture. Changes in the alignment of the lumbar spine appear according to standing and various sitting postures (Figure 1), which can be observed in X-ray images [26]. The model consists of various components. The thoracic and cervical vertebrae were modeled as a line, whereas the L1 to L5 lumbar vertebrae were modeled in detail, including the cortical bone, cancellous bone, posterior bone, intervertebral discs (composed of nucleus pulposus and annulus fibrosus), endplate, facet joint, sacrum, and pelvis (Figure 2). X-ray images were used to model the vertebral body for each posture, and ANSYS SpaceClaim software was used for 3D modeling correction (Figure 3).

### 2.2. Mesh and Material Properties for the FEM

In this study, the elastic physical properties of the vertebral body were applied because the stress was analyzed by applying loads within the elastic range that generally occur during activities at the level of daily living. The FEM analysis was performed using the static structural module of the ANSYS Workbench. The mesh used was a second-order tetrahedron (10 nodes). Small mesh sizes yield high analysis accuracy but require a long analysis time; likewise, for large mesh sizes, the analysis accuracy will be low and the required time will be short. After setting the mesh size of the entire model to 1–5 mm at 1 mm intervals, the von Mises stress was calculated. As a result, a 2 mm mesh size was selected for the FEM analysis because the stress difference between 1 and 2 mm was within 1.9%. The mesh size and material properties of the FEM are summarized in Table 1.

### 2.3. Loading and Boundary Conditions

Three types of load modes—flexion, lateral bending, and axial rotation—were applied to the FEM; a load of 300 N was applied to the head; and a moment of 10 N·m was applied along the flexion, lateral bending, and axial rotation movements of the lumbar spine (Figure 4). For the FEM analysis, the contact area between the pelvis and the femur was fixed. Because it is rather difficult to subdivide and apply contact conditions to components of the human body, we assumed that each component was a bonding contact. We then assessed these three types of loading on three components of the spine: the cortical bone, the annulus fibers, and the nucleus fibrosus of the intervertebral disc.

## 3. Results

### 3.1. Flexion Mode

Comparing the L1, L2, L3, L4, and L5 cortical bones during standing and erect sitting on a chair in the flexion mode, the von Mises stress in the erect sitting on a chair position increased by −25%, 14%, 10%, 1%, and 104%, respectively. Comparing standing with slumped sitting on a chair, the von Mises stress in the slumped sitting on a chair position increased by 48%, 262%, 365%, 282%, and 290%, respectively. Comparing standing with sitting on the floor, the stress while sitting increased by 65%, 125%, 97%, 86%, and 109%, respectively (Table 2). 

Comparing the annulus fibers (between L2-L1, L3-L2, L4-L3, L5-L4, and S-L5) in the standing and erect sitting on a chair position, the von Mises stress in the erect sitting on a chair position increased by −17%, 28%, 34%, 44%, and 14%, respectively. Comparing standing and slumped sitting on a chair, the von Mises stress in the slumped sitting position increased by 53%, 88%, 50%, −54%, and −57%, respectively. Comparing standing and sitting on the floor, the von Mises stress in the sitting position on the floor increased by 65%, 150%, 164%, 114%, and 52%, respectively (Table 2).

Comparing the L2-L1, L3-L2, L4-L3, L5-L4, and S-L5 nucleus pulposus in the standing and erect sitting on a chair position, the stress in the erect sitting on a chair position increased by −19%, 30%, 25%, 31%, and 3%, respectively. Comparing standing and slumped sitting on a chair, the von Mises stress in the slumped sitting position increased by 97%, 94%, 57%, −60%, and −74%, respectively. Comparing standing and sitting on the floor, the von Mises stress in the sitting position on the floor increased by 72%, 206%, 114%, 146%, and 57%, respectively (Table 2). The distribution map of von Mises stress in the cortical bone, annulus fiber, and nucleus pulposus during flexion motion in each posture can be confirmed in Figure 5.

### 3.2. Lateral Bending Mode 

Compared with the L1, L2, L3, L4, and L5 cortical bones in the standing and erect sitting on a chair positions in the lateral bending mode, the von Mises stress in the erect sitting on a chair position increased by −24%, 14%, 11%, 4%, and 66%, respectively. Comparing standing with slumped sitting on a chair, the von Mises stress in the slumped sitting on a chair position increased by 42%, 313%, 403%, 332%, and 203%, respectively. Comparing the standing position with sitting on the floor, the von Mises stress in the latter increased by 116%, 168%, 127%, 113%, and 59%, respectively (Table 3).

Comparing the L2-L1, L3-L2, L4-L3, L5-L4, and S-L5 annulus fibers in the standing and erect sitting on a chair positions, the von Mises stress in the erect sitting on a chair position increased by −12%, 36%, 12%, 19%, and 12%, respectively. Comparing standing and slumped sitting on a chair, the von Mises stress in the slumped sitting position increased by 53%, 102%, 45%, 0.9%, and 48%, respectively. Comparing standing and sitting on the floor, the von Mises stress in the sitting position increased by 62%, 149%, 111%, 103%, and 68%, respectively (Table 3).

Comparing the L2-L1, L3-L2, L4-L3, L5-L4, and S-L5 nucleus pulposus while standing and during erect sitting, the von Mises stress in the erect sitting on a chair position increased by −15%, 44%, 16%, 17%, and 198%, respectively. Comparing standing and slumped sitting on a chair, the von Mises stress in the slumped sitting position increased by 95%, 125%, 72%, −8%, and −57%, respectively. Comparing standing and sitting on the floor, the von Mises stress in the sitting position increased by 69%, 208%, 97%, 146%, and 73%, respectively (Table 3). The distribution map of von Mises stress in the cortical bone, annulus fiber, and nucleus pulposus during lateral bending motion in each posture can be confirmed in Figure 6.

### 3.3. Axial Rotation Mode

Comparing the L1, L2, L3, L4, and L5 cortical bones in the standing and erect sitting on a chair positions in the axial rotation mode, the von Mises stress in the erect sitting on a chair position increased by −22%, 6%, 10%, 10%, and 92%, respectively. Comparing the standing position with slumped sitting on a chair, the von Mises stress in the slumped sitting on a chair position increased by 43%, 327%, 487%, 446%, and 287%, respectively. Comparing the standing position with sitting on the floor, the von Mises stress in the latter position increased by 65%, 178%, 162%, 169%, and 105%, respectively (Table 4).

Comparing the L2-L1, L3-L2, L4-L3, L5-L4, and S-L5 annulus fibers in standing with erect sitting, the von Mises stress in the erect sitting on a chair position increased by −20%, 10%, 17%, 57%, and 11%, respectively. Comparing standing and sitting on the floor, the von Mises stress from sitting on the floor increased by 49%, 107%, 78%, −36%, and −45%, respectively. Comparing standing and sitting on the floor, the von Mises stress in the sitting position on the floor increased by 63%, 161%, 208%, 220%, and 73%, respectively (Table 4).

Comparing the L2-L1, L3-L2, L4-L3, L5-L4, and S-L5 nucleus pulposus in standing and erect sitting on a chair, the von Mises stress in the erect sitting on a chair position increased by −22%, 26%, 28%, 50%, and −0.1%, respectively. Comparing standing and sitting on the floor, the von Mises stress in the sitting position on the floor increased by 91%, 132%, 128%, −47%, and −57%, respectively. Comparing standing and sitting on the floor, the von Mises stress in the sitting position on the floor increased by 70%, 226%, 188%, 296%, and 80%, respectively (Table 4). The distribution map of von Mises stress in the cortical bone, annulus fiber, and nucleus pulposus during axial rotation motion in each posture can be confirmed in Figure 7.

### 3.4. Von Mises Stress According to the Three Load Modes

In the erect sitting and standing postures, there was no significant difference in the pressure distribution of the annulus fiber and nucleus pulposus, representing intradiscal pressure, according to the three movements. However, compared to the pressure of the standing posture, slumped sitting on a chair extensively increased the pressure on the nucleus pulposus and annulus fiber in all three movements. In sitting on the floor, the pressure on the nucleus pulposus and annulus fiber was greater in all three movements compared to that in slumped sitting on a chair. The pressure on the annulus fibers and nucleus pulposus in the lumbar spine increased the most during sitting on the floor. Therefore, it can be seen that sitting on the floor has a bad effect on the spine.

## 4. Discussion

In this study, we found that the von Mises stress values of the annulus fibers and nucleus pulposus in both the erect sitting on a chair posture and the standing posture were similar regardless of the movement, which suggests that there is no significant difference in the von Mises stress values of the annulus fibers and nucleus pulposus because the lumbar lordosis is not different between the two postures. This is similar to results comparing intervertebral disc pressures in the erect sitting on a chair posture and the standing posture in static conditions, where no significant differences were found [12,14]. We found no significant difference in the von Mises stress values of the annulus fibers and nucleus pulposus in the standing and erect sitting on a chair postures with or without movement. However, the von Mises stress values of the annulus fibers and nucleus pulposus in the slumped sitting on a chair and sitting on the floor postures were significantly different when compared to those for the standing posture. These results were similar to those of previous studies conducted in each posture in a static state [12,16,30,31]. Ultimately, we found that the von Mises stress of the annulus fibers and nucleus pulposus increased in the posture with reduced lumbar lordosis even in a static state and that the von Mises stress increased further in the posture with reduced lumbar lordosis when moving in each posture.

The cortical bone plays a vital role in the load-bearing capacity of the vertebral body. As age-related bone mass loss progresses, the load-carrying capacity of the cortical bone increases significantly [32], rendering the cortex more susceptible to fractures. Fractures of the cortical bone within the vertebral body generally become more prevalent with aging, wherein the deformation of the cortical bone proves to be more sensitive to aging than the nucleus pulposus, trabecular matrix, bone endplates, and posterior elements [33,34,35]. In our study, the von Mises stress on the cortical bone, annular fibrosus, and nucleus pulposus is reduced in postures with proper spinal alignment due to the adequate maintenance of lumbar lordosis. However, in postures like slumped sitting on a chair and sitting on the floor with reduced lumbar lordosis, the load on the cortical bone significantly increases compared to upright sitting and standing. If an individual continues to adopt a posture with reduced lumbar lordosis as they age, the deformation of the cortical bone will accelerate in addition to the deformation caused by aging. Consequently, the risk of cortical bone fractures within the vertebral body will escalate.

In East Asia, some populations spend more time sitting on the floor than sitting on a chair [36,37]. Compared with other sitting postures, the load values applied to the lumbar structure while sitting on the floor were different. In particular, we confirmed that the von Mises stress on the annulus fibrosus and nucleus pulposus of L5-L4 and S-L5 increased compared to that in slumped sitting on a chair. While sitting on the floor, lumbar lordosis and sacral slope are reduced compared with those while sitting on a chair in a bent-over position. The von Mises stress applied to the annulus fibrosus and nucleus pulposus further increased when flexion, lateral bending, and axial rotation were applied in a posture in which lumbar lordosis was reduced. These results are similar to the increased intradiscal pressure during physiological activities of flexion, extension, lateral bending, and axial rotation in the hypolordotic type of lumbar spine noted in Wang’s study [38]. In other words, the stress applied to the annulus fibrosus and nucleus pulposus of the lumbar spine increases during physiological activity in a posture where lumbar lordosis decreases. While sitting on the floor, the possibility of degeneration or prolapse of the lumbar spinal disc increases. Therefore, it is especially important to sit in a posture that maintains lumbar lordosis when there is back pain or a disease of the lumbar structure.

Lumbar degenerative kyphosis (LDK) refers to a sagittal misalignment of the lumbar spine caused by unique lifestyle habits like prolonged squatting in agricultural work and engaging in daily activities on the floor [39]. Previous studies have indicated that LDK is primarily prevalent in rural regions of East Asia, such as Korea and Japan [19,20,39]. Patients diagnosed with LDK typically exhibit extensive degenerative alterations in the lower lumbar intervertebral discs and posterior joints spanning from the L2 to S1 levels, alongside atrophy and fatty changes in the lumbar extensors [20]. LDK has garnered considerable attention through numerous publications in Korea and Japan [19,20,39]. In contrast, LDK has been infrequently documented in Western countries; however, within Asian nations, LDK ranks among the most prevalent spinal deformities, likely due to distinct lifestyles and work-related postures.

The etiology of LDK is attributed to degenerative changes within the discs, encompassing disc space narrowing, vacuum phenomena in the discs, and thickening of the posterior joints, which collectively lead to a loss of lumbar lordosis [40]. Consistent with our study findings, the continued adoption of farming or daily life practices with reduced lumbar lordosis, such as slumped sitting in chairs or sitting on the floor, will escalate the load on the intervertebral disc and cortical bone. This increased load can result in disc degeneration, further disc space narrowing, and acceleration of vacuum phenomena within the disc. Moreover, in elderly women affected by osteoporosis, the load on the cortical bone is also heightened, consequently amplifying the risk of vertebral body fractures.

A previous study suggested a benefit of lumbar lordosis, noting an association between reduced lumbar lordosis and increased disc degeneration [41,42]. Similarly, another study reported an inverse correlation between lumbar lordosis and intradiscal pressure of the lumbar spine [43,44]. Although Nachemson’s study did not explicitly establish lordosis as a determinant of intravertebral disc pressure, a careful examination revealed that intravertebral disc pressure increased at locations where lordosis was diminished [12]. An investigation into the relationship between lumbar lordosis and sitting highlighted that the most important factor for back pain caused by long-term sitting was the reduction in the trunk–thigh angle resulting from the flattening of the lumbar spine curve [45].

For individuals who spend the majority of their day sitting, our findings emphasize the importance of adopting proper sitting postures. The results of our study confirmed that the stress applied to the lumbar spine decreased in an erect sitting posture that maintains lumbar lordosis. Other studies have also indicated that the load on the lumbar spine is lower when sitting with support compared to sitting without support. This is due to the upper body’s weight being partially transferred to the chair’s back and armrests, consequently alleviating the load on the spine [16,46,47].

This study had some limitations. First, in the FEM, each vertebral structure was analyzed using fixed material properties. Second, our FEM did not encompass muscles, tendons, nerves, and ligaments around the lumbar spine, nor did it account for potential structural changes in these elements. The muscles surrounding the spine play a crucial role in stabilizing the spinal structures and alleviating stress on the spine [48,49]. Notably, the collective cross-sectional area of the numerous muscles encircling the spinal column far surpasses that of the spinal column itself. Furthermore, the muscles possess significantly larger lever arms in comparison to the intervertebral discs and ligaments. These muscles contribute to providing mechanical stability to the spinal column [48]. In a previous study, we observed that the von Mises stress within the vertebral body, intervertebral disc, and posterior column of the lumbar spine decreased as the volume of paraspinal muscles increased [50]. The inclusion of the multifidus muscle in FEM modeling can reduce the error of overestimating the compression loading of the lumbar spine in muscle-free simulations. It has been found that muscles which stabilize the lumbar spine, such as the multifidus muscle, can decrease the shear forces acting on the intervertebral discs by stabilizing the spine during lumbar spine movements [51]. Consequently, it can be predicted that the more active the lumbar spine stabilizing muscles are, the less load is placed on the lumbar spine. In addition, previous studies have examined muscle activation patterns responsible for spinal stabilization in different sitting postures and found that core muscles contributing to spinal stability were less activated in slumped sitting compared to erect sitting [52,53]. Applying these findings to our results, we interpret that in postures with reduced lumbar lordosis, such as slumped sitting in a chair, there is reduced activation of spinal stabilizing muscles and consequently increased stress on lumbar structures, including cortical bone, annulus fibers, and nucleus pulposus. Consequently, in the erect sitting posture on a chair, the activation of the core muscles that stabilize the spine increases, further reducing the stresses on the cortical bone, annulus fibers, and nucleus pulposus. Therefore, if an FEM study could be performed taking into account the activation of the muscles, we would expect to observe a greater difference in the stresses on the cortical bone, annulus fibers, and nucleus pulposus between erect sitting on a chair and slumped sitting on a chair than our experimental results indicate. Third, changes in geometric parameters, such as disc height, intervertebral disc cross-sectional area, nucleus size and location, fiber network orientation, or several fiber layers, can affect intervertebral disc mechanical behavior, suggesting that FEMs with different geometries may yield different results. In the future, further studies to compensate for the limitations are warranted for a more accurate analysis.

## 5. Conclusions

In conclusion, this study emphasizes that maintaining an erect sitting posture on a chair can reduce von Mises stress on the annulus fibrosus, nucleus pulposus, and cortical bone to a level similar to that of standing, which is especially important for modern individuals who spend a significant amount of time sitting. Additionally, the posture of sitting on the floor, common in East Asian cultures, leads to increased von Mises stresses in the annulus fibrosus, nucleus pulposus, and cortical bone when compared to the upright sitting posture on a chair. This posture seems to be associated with the development of lumbar degenerative kyphosis, frequently observed in East Asia. Adopting a lifestyle that involves sitting upright in a chair and maintaining proper lumbar lordosis will play a pivotal role in preventing various degenerative disc diseases and spinal deformities.

## Figures and Tables

**Figure 1 bioengineering-10-01051-f001:**
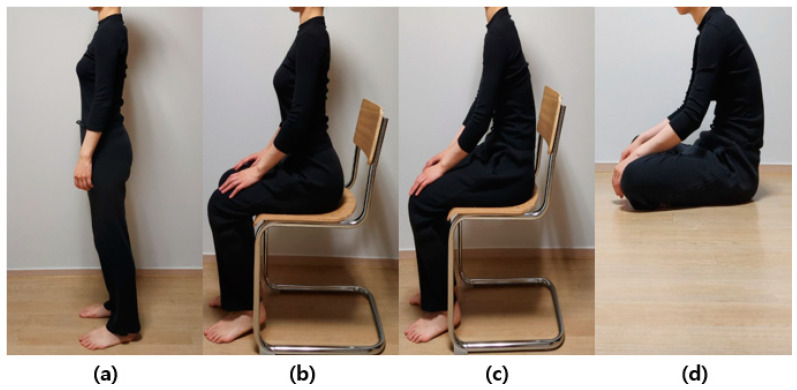
Lateral view of standing and various sitting postures observed in a healthy person: (**a**) standing, (**b**) erect sitting on a chair, (**c**) slumped sitting on a chair, and (**d**) sitting on the floor. Normally, the pelvis tilts backward during the transition to a sitting position, and the degree can vary depending on the sitting position. Lumbar lordosis decreases from (**a**–**d**).

**Figure 2 bioengineering-10-01051-f002:**
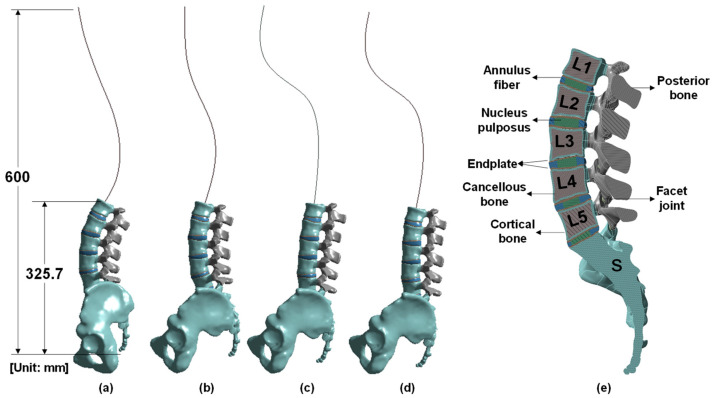
Vertebral body finite element model by posture and cross-section: (**a**) standing, (**b**) erect sitting on a chair, (**c**) slumped sitting on a chair, (**d**) sitting on the floor, and (**e**) cross-section of the lumbar spine.

**Figure 3 bioengineering-10-01051-f003:**
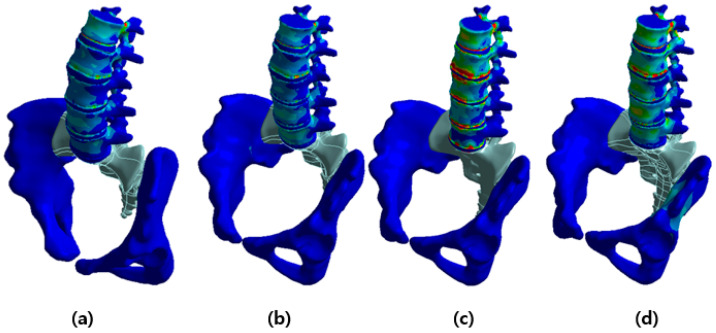
Changes in lumbar spine alignment according to each posture implemented through 3D modeling: (**a**) standing, (**b**) erect sitting on a chair, (**c**) slumped sitting on a chair, and (**d**) sitting on the floor.

**Figure 4 bioengineering-10-01051-f004:**
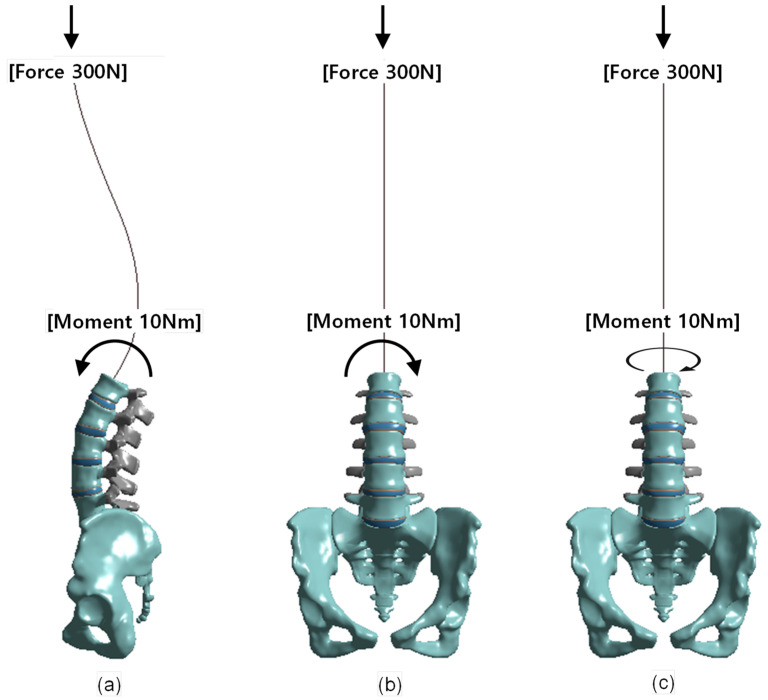
Three different motions are implemented by simultaneously applying axial force and moment. The axial force applied a load of 300 N to the upper surface of L1. A moment of 10 N·m was applied to trigger the motions of (**a**) flexion, (**b**) lateral bending, and (**c**) axial rotation.

**Figure 5 bioengineering-10-01051-f005:**
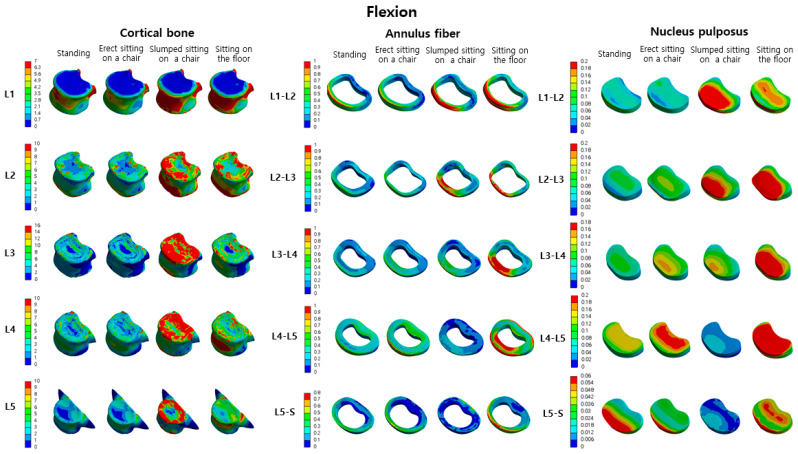
The von Mises stress distribution in the cortical bone, annulus fiber, and nucleus pulposus of the lumbar spine (L1–L5) in four postures (standing, erect sitting on a chair, slumped sitting on a chair, sitting on the floor) during flexion motion is displayed (unit: MPa).

**Figure 6 bioengineering-10-01051-f006:**
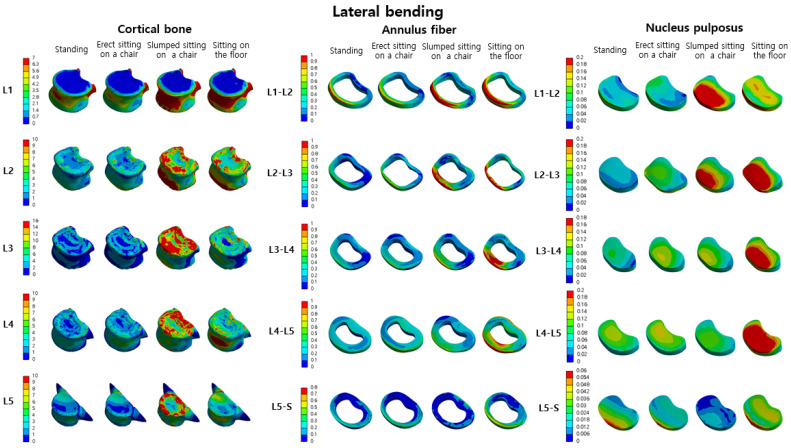
The von Mises stress distribution in the cortical bone, annulus fiber, and nucleus pulposus of the lumbar spine (L1–L5) in four postures (standing, erect sitting on a chair, slumped sitting on a chair, sitting on the floor) during lateral bending motion is displayed (unit: MPa).

**Figure 7 bioengineering-10-01051-f007:**
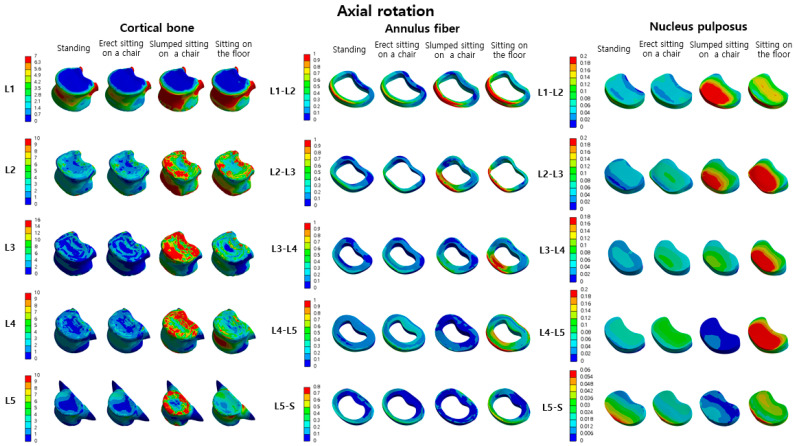
The von Mises stress distribution in the cortical bone, annulus fiber, and nucleus pulposus of the lumbar spine (L1–L5) in four postures (standing, erect sitting on a chair, slumped sitting on a chair, sitting on the floor) during axial rotation motion is displayed (unit: MPa).

**Table 1 bioengineering-10-01051-t001:** Information on mesh and material properties for the finite element model of the lumbar spine.

Component	Number of Nodes	Number of Elements	Elastic Modulus (MPa)	Poisson Ratio	Reference
Cortical bone	210,671	117,993	12,000	0.3	[28]
Cancellous bone	36,063	20,061	200	0.25	[28]
Posterior element	70,248	38,010	3500	0.25	[28]
End plate	44,100	21,009	1000	0.3	[28]
Nucleus pulposus	22,147	11,709	1	0.49	[28]
Annulus fiber	36,352	18,434	4.2	0.45	[28]
Facet joint	2222	990	11	0.4	[29]

**Table 2 bioengineering-10-01051-t002:** Results of von Mises stress in the cortical bone, annulus fiber, and nucleus pulposus of the lumbar spine according to flexion movement.

		Average (MPa)			
		Flexion			
		Standing	Erect Sitting	Slumped Sitting	Sitting on the Floor
Cortical Bone	L1	4.34	3.25	6.40	7.17
	L2	4.40	5.02	15.90	9.89
	L3	2.38	2.62	11.09	4.69
	L4	2.56	2.58	9.77	4.76
	L5	1.82	3.71	7.11	3.82
Annulus fiber	L2-L1	0.33	0.27	0.50	0.54
	L3-L2	0.22	0.29	0.42	0.56
	L4-L3	0.18	0.25	0.28	0.49
	L5-L4	0.31	0.45	0.15	0.67
	S-L5	0.16	0.19	0.07	0.25
Nucleus pulposus	L2-L1	0.08	0.06	0.15	0.13
	L3-L2	0.05	0.06	0.09	0.15
	L4-L3	0.05	0.06	0.07	0.10
	L5-L4	0.07	0.09	0.03	0.18
	S-L5	0.03	0.03	0.01	0.05

MPa, megapascal; S, sacrum.

**Table 3 bioengineering-10-01051-t003:** Results of von Mises stress in the cortical bone, annulus fiber, and nucleus pulposus of the lumbar spine according to lateral bending.

		Average (MPa)			
		Lateral Bending			
		Standing	Erect Sitting	Slumped Sitting	Sitting on the Floor
Cortical Bone	L1	4.38	3.33	6.22	7.18
	L2	3.12	3.55	12.87	8.34
	L3	1.71	1.89	8.58	3.88
	L4	1.73	1.80	7.46	3.68
	L5	1.92	3.18	5.82	3.06
Annulus fiber	L2-L1	0.33	0.29	0.51	0.54
	L3-L2	0.20	0.27	0.40	0.50
	L4-L3	0.20	0.22	0.29	0.42
	L5-L4	0.26	0.31	0.26	0.52
	S-L5	0.12	0.13	0.06	0.20
Nucleus pulposus	L2-L1	0.08	0.06	0.15	0.13
	L3-L2	0.04	0.06	0.10	0.13
	L4-L3	0.04	0.05	0.08	0.09
	L5-L4	0.06	0.07	0.05	0.14
	S-L5	0.02	0.06	0.01	0.04

MPa, megapascal; S, sacrum.

**Table 4 bioengineering-10-01051-t004:** Results of von Mises stress in the cortical bone, annulus fiber, and nucleus pulposus of the lumbar spine according to axial rotation.

		Average (MPa)			
		Axial Rotation			
		Standing	Erect Sitting	Slumped Sitting	Sitting on the Floor
Cortical Bone	L1	4.39	3.40	6.25	7.23
	L2	2.99	3.15	12.76	8.31
	L3	1.45	1.60	8.51	3.81
	L4	1.35	1.48	7.35	3.62
	L5	1.48	2.83	5.71	3.03
Annulus fiber	L2-L1	0.33	0.26	0.49	0.54
	L3-L2	0.19	0.20	0.38	0.49
	L4-L3	0.13	0.15	0.22	0.39
	L5-L4	0.15	0.24	0.10	0.49
	S-L5	0.12	0.13	0.06	0.20
Nucleus pulposus	L2-L1	0.08	0.06	0.15	0.13
	L3-L2	0.04	0.05	0.09	0.13
	L4-L3	0.03	0.04	0.06	0.08
	L5-L4	0.03	0.05	0.02	0.13
	S-L5	0.02	0.02	0.01	0.04

MPa, megapascal; S, sacrum.

## Data Availability

The datasets analyzed during the current study are available from the corresponding author upon reasonable request.

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
