# Peer review of "Biomechanical Effects of Different Sitting Postures and Physiologic Movements on the Lumbar Spine: A Finite Element Study"

_bioengineering, 2023, doi:10.3390/bioengineering10091051_

Round 1

Reviewer 1 Report

This study used finite element modeling (FEM) to explore the stress of lumbar spine under dynamic movements in different postures. The author provided insights and discussed in detail the pressure distribution of the lumbar vertebrae and surrounding structures under different modes. However, there are some comments for the authors.

(1) The core issues discussed behind this study are not clearly stated and there is no clear statement of what kind of problems this research will ultimately solve.

(2) Our first and primary concern lies in the novelty of this work, but the novelty issue has not been sufficiently highlighted in this current version of study. Based on the results at the end of this article, namely the distribution of pressure in the lumbar spine and surrounding tissues, can we go further and analyze something more in-depth based on this?

(3) This paper used the finite element method to analyze the pressure distribution of the lumbar spine. However, according to the whole process of this method, the paper seems to lack the process of comparing the results of finite element analysis with experimental data or theoretical expectations to verify the accuracy of the simulation.

(4) In the part of methodology, the selection of subjects is not explained in detail. It is necessary for the authors to provide such information more specifically to reflect the value of this paper.

(5) This study only investigated the distribution of pressure in the lumbar spine and surrounding structures. On this basis, can multiple force parameters be considered at the same time? Such as pressure, shear force, bending force, etc., to fully understand the comprehensive impact of different sitting postures under the stress of the lumbar spine.

(6) The conclusion and discussion part for this paper are relatively weak. It is advisable for the authors to compare and elaborate how the finite element method used in this paper supplements the previous researches.

 Moderate editing of English language required

Author Response

Dear Reviewer #1

Response to Reviewer 1 Comments

This study used finite element modeling (FEM) to explore the stress of lumbar spine under dynamic movements in different postures. The author provided insights and discussed in detail the pressure distribution of the lumbar vertebrae and surrounding structures under different modes. However, there are some comments for the authors.

Response :

Thank you very much for your kind letter and comments concerning our manuscript entitled

“Biomechanical Effects of Different Sitting Postures and Physiologic Movements on the Lumbar Spine: A Finite Element Study”

The comments were valuable and very helpful in critically revising and improving our paper. We have discussed the comments carefully and have made corresponding corrections which we hope will be met with approval. The following are responses to your comments

Comment 1)

The core issues discussed behind this study are not clearly stated and there is no clear statement of what kind of problems this research will ultimately solve.

Response 1  : Thank you for your valuable feedback. We have meticulously reviewed your comments and acknowledge that the key focal point of our study was not adequately clarified in the introduction, as you rightly pointed out. While numerous studies have endeavored to gauge the pressure on spinal structures in relation to posture, these investigations typically assessed pressure within each posture in a static state. Nonetheless, it is important to recognize that individuals do not maintain fixed positions while standing or sitting; rather, they engage in a variety of dynamic movements. The distinctive aspect of our study lies in our computation of spinal structure loads during various movements within each posture. This approach allows us to ascertain the optimal posture for minimizing stress on spinal structures, highlighting its significance. Additionally, our interest in the seated-on-the-floor posture, prevalent among East Asian cultures, stemmed from our observation of its potential to contribute to lumbar degenerative kyphosis. Consequently, we conducted a comprehensive analysis of spinal structure pressures during flexion, lateral bending, and axial rotation across four distinct postures: standing, erect sitting on a chair, slumped sitting on a chair, and sitting on the floor. The following sentences and references have been added for detailed explanation.

The following sentences and references have been added for detailed explanation.

Page 2, Introduction, Line 58-79

Calculations of intradiscal pressure and lumbar spinal load are widely used to prevent spinal disorders by forming the basis of clinical advice to promote correct sitting posture [11]. Many studies have aimed to measure the pressure distribution within the lumbar spine across various postures. These studies typically assessed the lumbar spine pressure in a static and immobile state within each position[12-16]. However, individuals commonly spend substantial periods standing or sitting, engaging in movements such as flexion, lateral bending, and axial rotation. Hence, comprehending the stress on the lumbar spine during diverse movements within each position, as opposed to only in static postures, is crucial for understanding actual lumbar spine stress.

It is estimated that adults in the United States spend over eight hours per day seated, illustrating that a significant portion of contemporary individuals conducts their daily activities while seated—working, eating, and engaging in leisure[17]. Furthermore, adopting a cross-legged sitting position on the floor is a prevalent practice in Asian cultures. Lumbar degenerative kyphosis(LDK) is often observed among elderly women in rural Asian regions who primarily adopt the sitting on the floor posture[18, 19]. Notably, it's understood that the sitting on the floor posture linked to LDK[18, 20].

Therefore, this study investigates the pressure exerted on spinal structures during three movements (flexion, lateral bending, and axial rotation) in a total of four postures (standing, sitting upright in a chair, slouching in a chair, and sitting on the floor), including the addition of sitting on the floor. The study utilizes finite element model(FEM) analysis, and its primary objective is to determine the optimal posture for minimizing stress on the spinal structures during movement.

  1. Araújo, A., et al., On the experimental intradiscal pressure measurement techniques: a review. New Trends in Mechanism and Machine Science, 2015: p. 243-250.
  2. Nachemson, A., The load on lumbar disks in different positions of the body. Clinical Orthopaedics and Related Research (1976-2007), 1966. 45: p. 107-122.
  3. Goto, K., et al., Mechanical analysis of the lumbar vertebrae in a three-dimensional finite element method model in which intradiscal pressure in the nucleus pulposus was used to establish the model. Journal of orthopaedic science, 2002. 7(2): p. 243-246.
  4. Li, J.-Q., et al., Comparison of in vivo intradiscal pressure between sitting and standing in human lumbar spine: a systematic review and meta-analysis. Life, 2022. 12(3): p. 457.
  5. Nachemson, A. and J.M. Morris, In Vivo Measurements of Intradiscal Pressure. Discometry, a Method for the Determination of Pressure in the Lower Lumbar Discs. J Bone Joint Surg Am, 1964. 46: p. 1077-92.
  6. Wilke, H.J., et al., New in vivo measurements of pressures in the intervertebral disc in daily life. Spine, 1999. 24(8): p. 755-762.
  7. Matthews, C.E., et al., Amount of time spent in sedentary behaviors in the United States, 2003-2004. Am J Epidemiol, 2008. 167(7): p. 875-81.
  8. Hong, J.-H., et al., Is the agricultural work a risk factor for Koreans elderly spinal sagittal imbalance? Journal of Korean Neurosurgical Society, 2020. 63(5): p. 623-630.
  9. Lee, S.-H., et al., Sagittal decompensation after corrective osteotomy for lumbar degenerative kyphosis: classification and risk factors. Spine, 2011. 36(8): p. E538-E544.
  10. Lee, C.-S., Y.-T. Kim, and E. Kim, Clinical study of lumbar degenerative kyphosis. Journal of Korean Society of Spine Surgery, 1997. 4(1): p. 27-35.

Comment 2)

Our first and primary concern lies in the novelty of this work, but the novelty issue has not been sufficiently highlighted in this current version of study. Based on the results at the end of this article, namely the distribution of pressure in the lumbar spine and surrounding tissues, can we go further and analyze something more in-depth based on this?

Response 2 : Thank you for your valuable feedback. We have conducted a thorough review of your comments. In order to underscore the novelty of our study, we have taken care to emphasize in the introduction that we examined the pressure on spinal structures within various postures, considering not just static positions as mentioned in Comment 1), but also incorporating dynamic movements.

Furthermore, we have strived to delve deeper into the final result of our study, which revealed heightened von Mises stresses in the annulus fibrosus, nucleus pulposus, and cortical bone during slumped sitting on a chair and sitting on the floor, in comparison to the standing and erect sitting on a chair positions.

  • The cortical bone's significance in vertebral body load-bearing is crucial. With advancing age-related bone mass loss, cortical bone load-bearing capacity notably rises, increasing its susceptibility to fractures. Such fractures become more common as individuals age, particularly affecting cortical bone deformation compared to other elements like the nucleus pulposus, trabecular matrix, bone endplates, and posterior elements. Our study indicates that proper spinal alignment in postures preserves lumbar lordosis, resulting in reduced stress on the cortical bone, annular fibrosus, and nucleus pulposus. However, postures like slumped sitting or sitting on the floor, which diminish lumbar lordosis, substantially heighten cortical bone stress relative to upright sitting and standing. If an individual persists in a reduced lumbar lordosis posture as they age, the cortical bone's deformation accelerates alongside age-related changes, elevating the risk of vertebral body cortical bone fractures.
  • Lumbar degenerative kyphosis (LDK) involves sagittal misalignment of the lumbar spine due to specific lifestyle practices like prolonged squatting during agricultural work and daily activities on the floor. This condition is mainly observed in rural areas of East Asia, such as Korea and Japan. Patients with LDK display significant degenerative changes in lower lumbar intervertebral discs and posterior joints (L2 to S1 levels), alongside atrophy and fatty changes in lumbar extensor muscles. While LDK is extensively discussed in publications from Korea and Japan, it's less documented in Western countries. However, within Asian nations, LDK is a prevalent spinal deformity, likely due to distinctive lifestyles and work-related postures. LDK's origin lies in degenerative disc changes, including disc space narrowing, vacuum phenomena in the discs, and thickened posterior joints, leading to a loss of lumbar lordosis. Consistent with our study, continuing practices with reduced lumbar lordosis, like slumped sitting or sitting on the floor, can increase intervertebral disc and cortical bone load. This intensified load contributes to disc degeneration, further disc space narrowing, and increased vacuum phenomena. In elderly women with osteoporosis, this also raises cortical bone load, heightening the risk of vertebral body fractures.

The following sentences and references have been added for detailed explanation.

Page 11, Discussion, Line 292-305

The cortical bone plays a vital role in the load-bearing capacity of the vertebral body. As age-related bone mass loss progresses, the load-carrying capacity of the cortical bone increases significantly[34], rendering the cortex more susceptible to fractures. Fractures of the cortical bone within the vertebral body generally become more prevalent with aging, wherein the deformation of the cortical bone proves to be more sensitive to aging than the nucleus pulposus, trabecular matrix, bone endplates, and posterior elements[35, 36]. In our study, the stress on the cortical bone, annular fibrosus, and nucleus pulposus is reduced in postures with proper spinal alignment due to the adequate maintenance of lumbar lordosis. However, in postures like slumped sitting on a chair and sitting on the floor with reduced lumbar lordosis, the load on the cortical bone significantly increases compared to upright sitting and standing. If an individual continues to adopt a posture with reduced lumbar lordosis as they age, the deformation of the cortical bone will accelerate in addition to the deformation caused by aging. Consequently, the risk of cortical bone fractures within the vertebral body will escalate.

  1. Cao, K.D., M.J. Grimm, and K.-H. Yang, Load sharing within a human lumbar vertebral body using the finite element method. Spine, 2001. 26(12): p. e253-e260.
  2. Lu, Y., et al., Strain changes on the cortical shell of vertebral bodies due to spine ageing: A parametric study using a finite element model evaluated by strain measurements. Proceedings of the Institution of Mechanical Engineers, Part H: Journal of Engineering in Medicine, 2013. 227(12): p. 1265-1274.
  3. Christiansen, B.A., et al., Mechanical contributions of the cortical and trabecular compartments contribute to differences in age‐related changes in vertebral body strength in men and women assessed by QCT‐based finite element analysis. Journal of Bone and Mineral Research, 2011. 26(5): p. 974-983.

Page 12, Discussion, Line 322-342

Lumbar degenerative kyphosis (LDK) refers to a sagittal misalignment of the lumbar spine caused by unique lifestyle habits like prolonged squatting in agricultural work and engaging in daily activities on the floor[40]. Previous studies have indicated that LDK is primarily prevalent in rural regions of East Asia, such as Korea and Japan[19, 20, 40]. Patients diagnosed with LDK typically exhibit extensive degenerative alterations in the lower lumbar intervertebral discs and posterior joints spanning from the L2 to S1 levels, alongside atrophy and fatty changes in the lumbar extensors[20]. LDK has garnered considerable attention through numerous publications in Korea and Japan[19, 20, 40]. In contrast, LDK has been infrequently documented in Western countries; however, within Asian nations, LDK ranks among the most prevalent spinal deformities, likely due to distinct lifestyles and work-related postures.

The etiology of LDK is attributed to degenerative changes within the discs, encompassing disc space narrowing, vacuum phenomena in the discs, and thickening of the posterior joints, which collectively lead to a loss of lumbar lordosis[41]. Consistent with our study findings, the continued adoption of farming or daily life practices with reduced lumbar lordosis, such as slumped sitting in chairs or sitting on the floor, will escalate the load on the intervertebral disc and cortical bone. This increased load can result in disc degeneration, further disc space narrowing, and acceleration of vacuum phenomena within the disc. Moreover, in elderly women affected by osteoporosis, the load on the cortical bone is also heightened, consequently amplifying the risk of vertebral body fractures.

  1. Lee, S.-H., et al., Sagittal decompensation after corrective osteotomy for lumbar degenerative kyphosis: classification and risk factors. Spine, 2011. 36(8): p. E538-E544.
  2. Lee, C.-S., Y.-T. Kim, and E. Kim, Clinical study of lumbar degenerative kyphosis. Journal of Korean Society of Spine Surgery, 1997. 4(1): p. 27-35.
  3. Takemitsu, Y., et al., Lumbar degenerative kyphosis. Clinical, radiological and epidemiological studies. Spine, 1988. 13(11): p. 1317-1326.
  4. Jang, J.-S., et al., Lumbar degenerative kyphosis: radiologic analysis and classifications. Spine, 2007. 32(24): p. 2694-2699.

Comment 3)

This paper used the finite element method to analyze the pressure distribution of the lumbar spine. However, according to the whole process of this method, the paper seems to lack the process of comparing the results of finite element analysis with experimental data or theoretical expectations to verify the accuracy of the simulation.

Response 3 : Thank you for your valuable feedback. We have conducted a thorough review of your comments. The finite element model of the vertebral body used in this study is based on data collected by KISTI* and used by agreement. Based on this, we have published papers using the same finite element model as the vertebral body model used in this study [Jeong et al. 2022, Kang et al. 2022, Song et al. 2022]. In addition, the size of the vertebral body finite element model used in this study was compared with experimental data in the existing literature to prove its appropriateness [Wolf et al. 2001]. In other words, the vertebral body finite element model for each posture (standing, erect sitting on the chair, slumped sitting on the chair, and sitting on the floor) was built using the verified vertebral body finite element model, so the vertebral body finite element model was sufficiently verified.

*KISTI provided the Korean human information (such as CT, MR, serially sectioned image, segmented image and 3D image) since 2000 and many kinds of Korean human data were prepared and serviced to the users in domestic and abroad.

Jeong et al., 2022, Biomechanical Effect of Disc Height on the Components of the Lumbar Column at the Same Axial Load: A Finite-Element Study, Journal of Healthcare Engineering 2022.

Kang et al., 2022, Analysis of the physiological load on lumbar vertebrae in patients with osteoporosis: A finite-element study, Scientific Reports 12 (1).

Song et al., 2022, Effects of location and volume of intraosseous cement on adjacent level of osteoporotic spine undergoing kyphoplasty: Finite element analysis, World Neurosurgery 162.

Wolf et al., 2001, Morphometric study of the human lumbar spine for operation–workspace specifications, Spine 26.

Comment 4)

In the part of methodology, the selection of subjects is not explained in detail. It is necessary for the authors to provide such information more specifically to reflect the value of this paper.

Response 4 :  Thank you for your valuable feedback. We've carefully reviewed your comment. In our study, there are no distinct subjects; rather, the finite element model was purposefully crafted to accurately represent men in their mid-30s, with an average height of 175 cm. Digital data of the human body were obtained from the Korea Institute of Science and Technology Information (KISTI) and utilized under a formal agreement. Notably, Institutional Review Board (IRB) approval was not a requisite in this scenario. Since the year 2000, KISTI has been furnishing Korean human information, encompassing CT scans, MRIs, serially sectioned images, segmented images, 3D images, and X-ray images. A diverse array of Korean human datasets has been meticulously curated and made accessible to both domestic and international users. To substantiate its appropriateness, we compared the size of the vertebral body finite element model employed in our study with experimental data in the existing literature.

The following sentences and references have been added for detailed explanation.

Page 2-3, Method, Line 82-98

A 3D finite element model(FEM) centered on the lumbar spine was constructed for this study[21, 22]. The finite element model was specifically created to represent men in their mid-30s, with an average height of 175 cm. Digital data of the human body were acquired from the Korea Institute of Science and Technology Information (KISTI) and used under an agreement. No Institutional Review Board (IRB) approval was required in this case.

KISTI has been providing Korean human information, including CT scans, MRIs, serially sectioned images, segmented images, and 3D images, Xray imange since the year 2000. A variety of Korean human datasets have been prepared and made available to users both domestically and internationally. The size of the vertebral body finite element model used in this study was compared with experimental data in the existing literature to prove its appropriateness[23].

The vertebral body for each posture was modeled using X-ray images from a previous work and a previously created FEM [24-27]. Data on lumbar lordosis and pelvic parameters important for spine alignment in each posture are based on a previous study[25, 27]. In particular, the size of the constructed lumbar spine model was valid within the error range when compared with the actual measurement results of a previous study [28].

  1. Jain, P. and M.R. Khan, Comparison of novel stabilisation device with various stabilisation approaches: A finite element based biomechanical analysis. The International Journal of Artificial Organs, 2022. 45(5): p. 514-522.
  2. Jain, P., et al., Biomechanics of spinal implants—a review. Biomedical Physics & Engineering Express, 2020. 6(4): p. 042002.
  3. Wolf, A., et al., Morphometric study of the human lumbar spine for operation–workspace specifications. Spine, 2001. 26(22): p. 2472-2477.
  4. Patwardhan, A.G., et al., Loading of the lumbar spine during transition from standing to sitting: effect of fusion versus motion preservation at L4–L5 and L5–S1. The Spine Journal, 2021. 21(4): p. 708-719.
  5. Bae, J.S., et al., A comparison study on the change in lumbar lordosis when standing, sitting on a chair, and sitting on the floor in normal individuals. Journal of Korean Neurosurgical Society, 2012. 51(1): p. 20-23.
  6. Song, S.-Y., et al., Effects of Location and Volume of Intraosseous Cement on Adjacent Level of Osteoporotic Spine Undergoing Kyphoplasty: Finite Element Analysis. World Neurosurgery, 2022.
  7. Cho, I.Y., et al., The effect of standing and different sitting positions on lumbar lordosis: radiographic study of 30 healthy volunteers. Asian Spine Journal, 2015. 9(5): p. 762.
  8. Kang, S., et al., Analysis of the physiological load on lumbar vertebrae in patients with osteoporosis: a finite-element study. Scientific Reports, 2022. 12(1): p. 1-14.

Comment 5)

This study only investigated the distribution of pressure in the lumbar spine and surrounding structures. On this basis, can multiple force parameters be considered at the same time? Such as pressure, shear force, bending force, etc., to fully understand the comprehensive impact of different sitting postures under the stress of the lumbar spine.

Response 5 : Thank you for your valuable feedback. We've carefully reviewed your comment. As you mentioned, it is necessary to consider all the stresses (load applied per unit area) on the vertebral body structure (bone, disc). In this study, we calculated the stresses on the vertebral body structure by considering the von Mises stress, which includes the stresses caused by both axial and shear loads, and the results are shown in Table 2-4 and Figure 5. The von Mises stress is the stress caused by the composite load components expressed in the figure and equation below. Therefore, the results of this study already represent the stress analysis caused by the composite load.

where, σv is von mises stress

σxyz are normal stress and τxy, τyz, τzx are shear stress in each axis direction

Comment 6)

The conclusion and discussion part for this paper are relatively weak.

It is advisable for the authors to compare and elaborate how the finite element method used in this paper supplements the previous researches.

Response 6 : Thank you for your valuable feedback. We have thoroughly reviewed your comments and have revised and enhanced the discussion and conclusion sections to incorporate your input.

The following sentences and references have been added for detailed explanation.

Page 11, Discussion, Line 276-291

< Sentence before improvement>

In this study, we confirmed that a large amount of von Mises stress is applied to the spinal structure in postures where lumbar lordosis is reduced, such as slumped sitting on a chair or sitting on the floor. This result is in agreement with previous studies. However, in previous studies, intradiscal pressure was measured by inserting a pressure sensor into pig vertebrae or directly into a human disc. These pioneering findings form the basis of our current knowledge of the in vivo loading conditions on the human spine but indicate that further studies are needed to better understand the mechanical loading of the lumbar spine. In this study, FEM analysis based on the physical properties of the human body was used.

We found that the stress values on the annulus fiber and nucleus pulposus were similar in both standing and erect sitting on a chair postures regardless of movement, likely because lumbar lordosis does not differ between these two postures, suggesting no significant difference in pressure on the spine and discs. Previous studies also showed conflicting results in the comparison of intravertebral disc pressure between an erect sitting posture on a chair and a standing posture, but no significant difference was found.

Slumped sitting on a chair and sitting on the floor showed that a large pressure difference was applied to the spine upon sitting compared to that with standing, in line with the findings of several studies. In addition, there was a marked pressure difference in slumped sitting when compared to that during erect sitting on a chair. Our results con-firmed the findings of previous studies, indicating that as lumbar lordosis decreased, the pressure on the intervertebral disc increased. This relationship was observed even during sitting, likely due to variations in waist posture. Our study provides evidence that maintaining an erect sitting posture can reduce the burden on the back, which is particularly important in modern society, where many people spend most of their day sitting and are at risk of back pain and related conditions.

< Sentence after improvement>

In this study, we found that the stress values of the annulus fibers and nucleus pulposus in both the erect sitting on a chair posture and the standing posture were similar regardless of the movement, which suggests that there is no significant difference in the stress values of the annulus fibers and nucleus pulposus because the lumbar lordosis is not different between the two postures. This is similar to results comparing intervertebral disc pressures in the erect sitting on a chair posture and the standing posture in static conditions, where no significant differences were found[12, 24, 31]. We found no significant difference in the stress values of the annulus fibers and nucleus pulposus in the standing and erect sitting on a chair postures with or without movement. However, the stress values of the annulus fibers and nucleus pulposus in the slumped sitting on a chair and sitting on the floor postures was significantly different when compared to the standing posture. These results were similar to previous studies conducted in each posture in a static state[12, 31-33]. Ultimately, we found that the stress of the annulus fibers and nucleus pulposus increased in the posture with reduced lumbar lordosis even in a static state, and that the stress increased further in the posture with reduced lumbar lordosis when moving in each posture.

  1. Nachemson, A., The load on lumbar disks in different positions of the body. Clinical Orthopaedics and Related Research (1976-2007), 1966. 45: p. 107-122.
  2. Li, J.-Q., et al., Comparison of in vivo intradiscal pressure between sitting and standing in human lumbar spine: a systematic review and meta-analysis. Life, 2022. 12(3): p. 457.
  3. Wilke, H.J., et al., New in vivo measurements of pressures in the intervertebral disc in daily life. Spine (Phila Pa 1976), 1999. 24(8): p. 755-62.
  4. Andersson, G., R. Ortengren, and A. Nachemson, Quantitative studies of the load on the back in different working-postures. Scandinavian journal of rehabilitation medicine. Supplement, 1978. 6: p. 173-181.
  5. Lord, M.J., et al., Lumbar lordosis. Effects of sitting and standing. Spine (Phila Pa 1976), 1997. 22(21): p. 2571-4.

Page 12, Discussion, Line 343-358

< Sentence before improvement>

A benefit of lumbar lordosis was suggested by a previous study, which noted an as-sociation between reduced lumbar lordosis and increased disc degeneration at S-L5 [32]. Another study also reported an inverse relationship between lordosis and intradiscal pressure [33]. Although Nachemson’s study did not specifically identify lordosis as a de-terminant of intravertebral pressure, a careful review showed that intravertebral pressure increased at locations where lordosis was reduced [26]. A study of the relationship be-tween lumbar lordosis and sitting found that the most important factor for back pain caused by long-term sitting was a reduction in the trunk-thigh angle resulting from the flattening of the lumbar spine curve [34].

For those who spend most of their day sitting, our results indicate that proper sitting postures should be considered. The results of our study confirmed that the pressure ap-plied to the lumbar spine was reduced in the erect sitting posture while maintaining lumbar lordosis. In other studies, the load applied to the lumbar spine was lower when sitting with support than when sitting without support because the weight of the upper body is partially transferred to the chair back and armrests, reducing the load on the spine [27,35,36].

< Sentence after improvement>

A previous study suggested a benefit of lumbar lordosis, noting an association between reduced lumbar lordosis and increased disc degeneration at S-L5[42]. Similarly, Another study reported an inverse correlation between lordosis and intradiscal pressure [43]. Although Nachemson’s study did not explicitly establish lordosis as a determinant of intravertebral pressure, a careful examination revealed that intravertebral pressure increased at locations where lordosis was diminished [12]. An investigation into the relationship between lumbar lordosis and sitting highlighted that the most important factor for back pain caused by long-term sitting was the reduction in the trunk-thigh angle resulting from the flattening of the lumbar spine curve [44].

For individuals who spend a majority of their day sitting, our findings emphasize the importance of adopting proper sitting postures. The results of our study confirmed that the stress applied to the lumbar spine decreased in an erect sitting posture that maintains lumbar lordosis. Other studies have also indicated that the load on the lumbar spine is lower when sitting with support compared to sitting without support. This is due to the upper body's weight being partially transferred to the chair's back and armrests, consequently alleviating the load on the spine [16, 45, 46].

  1. Nachemson, A., The load on lumbar disks in different positions of the body. Clinical Orthopaedics and Related Research (1976-2007), 1966. 45: p. 107-122.
  2. Wilke, H.J., et al., New in vivo measurements of pressures in the intervertebral disc in daily life. Spine, 1999. 24(8): p. 755-762.
  3. Farfan, H., R. Huberdeau, and H. Dubow, Lumbar intervertebral disc degeneration: the influence of geometrical features on the pattern of disc degeneration—a post mortem study. JBJS, 1972. 54(3): p. 492-510.
  4. Adams, M. and W. Hutton, The effect of posture on the lumbar spine. The Journal of bone and joint surgery. British volume, 1985. 67(4): p. 625-629.
  5. Williams, M.M., et al., A comparison of the effects of two sitting postures on back and referred pain. Spine, 1991. 16(10): p. 1185-1191.
  6. Aota, Y., et al., Effectiveness of a lumbar support continuous passive motion device in the prevention of low back pain during prolonged sitting. Spine (Phila Pa 1976), 2007. 32(23): p. E674-7.
  7. Moriguchi, C.S., T.O. Sato, and H. Coury, An Instrumented Workstation to Evaluate Weight-Bearing Distribution in the Sitting Posture. Saf Health Work, 2019. 10(3): p. 314-320.

Page 13, Conclusion, Line 385-393

< Sentence before improvement>

In conclusion, this study highlights that maintaining an erect sitting posture on a chair can effectively reduce lumbar spine pressure to a level comparable to standing for individuals who spend a significant amount of time sitting. Moreover, it is crucial to avoid sitting on the floor, a common practice in Asian culture, as it substantially increases intra-discal pressure. Instead, adopting an erect sitting position on a chair and making lifestyle changes to promote proper lumbar lordosis maintenance will play a pivotal role in preventing various degenerative disc diseases.

< Sentence after improvement>

In conclusion, this study emphasizes that maintaining an erect sitting posture on a chair can reduce von Mises stress on the annulus fibrosus, nucleus pulposus, and cortical bone to a level similar to that of standing, especially important for modern individuals who spend a significant amount of time sitting. Additionally, the posture of sitting on the floor, common in East Asian cultures, leads to increased von Mises stresses in the annulus fibrosus, nucleus pulposus, and cortical bone when compared to the upright sitting posture on a chair. This posture seems to be associated with the development of lumbar degenerative kyphosis, frequently observed in East Asia. Adopting a lifestyle that involves sitting upright in a chair and maintaining proper lumbar lordosis will play a pivotal role in preventing various degenerative disc diseases and spinal deformities.

Reviewer 2 Report

Very interesting article on a previously unexplained topic.

Applies to a Finite Element Study  of Biomechanical Effects of Different Sitting Postures and Physiologic Movements on the Lumbar Spine.

Abstract: Well written, readable.

Introduction:

Please clearly describe why this research topic was chosen.

Please describe in detail why this study is so important and unique.

Please add a research hypothesis.

Material and methods:

Well and accurately described, legible figures.

Results:

Well and accurately described, legible figures and tables.

Discussion:

Please clearly describe why this research topic was chosen.

Please describe in detail why this study is so important and unique.

Please add a practical application for doctors and physiotherapists.

Author Response

Dear Reviewer #2

Response to Reviewer 2 Comments

Very interesting article on a previously unexplained topic.

Applies to a Finite Element Study of Biomechanical Effects of Different Sitting Postures and Physiologic Movements on the Lumbar Spine.

Response :

Thank you very much for your kind letter and comments concerning our manuscript entitled

“Biomechanical Effects of Different Sitting Postures and Physiologic Movements on the Lumbar Spine: A Finite Element Study”

The comments were valuable and very helpful in critically revising and improving our paper. We have discussed the comments carefully and have made corresponding corrections which we hope will be met with approval. The following are responses to your comments

Comment 1)

Introduction:

Please clearly describe why this research topic was chosen.

Please describe in detail why this study is so important and unique.

Please add a research hypothesis.

Response 1  : Thank you for your valuable feedback. We have meticulously reviewed your comments and acknowledge that the key focal point of our study was not adequately clarified in the introduction, as you rightly pointed out. While numerous studies have endeavored to gauge the pressure on spinal structures in relation to posture, these investigations typically assessed pressure within each posture in a static state. Nonetheless, it is important to recognize that individuals do not maintain fixed positions while standing or sitting; rather, they engage in a variety of dynamic movements. The distinctive aspect of our study lies in our computation of spinal structure loads during various movements within each posture. This approach allows us to ascertain the optimal posture for minimizing stress on spinal structures, highlighting its significance. Additionally, our interest in the seated-on-the-floor posture, prevalent among East Asian cultures, stemmed from our observation of its potential to contribute to lumbar degenerative kyphosis. Consequently, we conducted a comprehensive analysis of spinal structure pressures during flexion, lateral bending, and axial rotation across four distinct postures: standing, erect sitting on a chair, slumped sitting on a chair, and sitting on the floor. The following sentences and references have been added for detailed explanation.

The following sentences and references have been added for detailed explanation.

Page 2, Introduction, Line 58-79

Calculations of intradiscal pressure and lumbar spinal load are widely used to prevent spinal disorders by forming the basis of clinical advice to promote correct sitting posture [11]. Many studies have aimed to measure the pressure distribution within the lumbar spine across various postures. These studies typically assessed the lumbar spine pressure in a static and immobile state within each position[12-16]. However, individuals commonly spend substantial periods standing or sitting, engaging in movements such as flexion, lateral bending, and axial rotation. Hence, comprehending the stress on the lumbar spine during diverse movements within each position, as opposed to only in static postures, is crucial for understanding actual lumbar spine stress.

It is estimated that adults in the United States spend over eight hours per day seated, illustrating that a significant portion of contemporary individuals conducts their daily activities while seated—working, eating, and engaging in leisure[17]. Furthermore, adopting a cross-legged sitting position on the floor is a prevalent practice in Asian cultures. Lumbar degenerative kyphosis(LDK) is often observed among elderly women in rural Asian regions who primarily adopt the sitting on the floor posture[18, 19]. Notably, it's understood that the sitting on the floor posture linked to LDK[18, 20].

Therefore, this study investigates the pressure exerted on spinal structures during three movements (flexion, lateral bending, and axial rotation) in a total of four postures (standing, sitting upright in a chair, slouching in a chair, and sitting on the floor), including the addition of sitting on the floor. The study utilizes finite element model(FEM) analysis, and its primary objective is to determine the optimal posture for minimizing stress on the spinal structures during movement.

  1. Araújo, A., et al., On the experimental intradiscal pressure measurement techniques: a review. New Trends in Mechanism and Machine Science, 2015: p. 243-250.
  2. Nachemson, A., The load on lumbar disks in different positions of the body. Clinical Orthopaedics and Related Research (1976-2007), 1966. 45: p. 107-122.
  3. Goto, K., et al., Mechanical analysis of the lumbar vertebrae in a three-dimensional finite element method model in which intradiscal pressure in the nucleus pulposus was used to establish the model. Journal of orthopaedic science, 2002. 7(2): p. 243-246.
  4. Li, J.-Q., et al., Comparison of in vivo intradiscal pressure between sitting and standing in human lumbar spine: a systematic review and meta-analysis. Life, 2022. 12(3): p. 457.
  5. Nachemson, A. and J.M. Morris, In Vivo Measurements of Intradiscal Pressure. Discometry, a Method for the Determination of Pressure in the Lower Lumbar Discs. J Bone Joint Surg Am, 1964. 46: p. 1077-92.
  6. Wilke, H.J., et al., New in vivo measurements of pressures in the intervertebral disc in daily life. Spine, 1999. 24(8): p. 755-762.
  7. Matthews, C.E., et al., Amount of time spent in sedentary behaviors in the United States, 2003-2004. Am J Epidemiol, 2008. 167(7): p. 875-81.
  8. Hong, J.-H., et al., Is the agricultural work a risk factor for Koreans elderly spinal sagittal imbalance? Journal of Korean Neurosurgical Society, 2020. 63(5): p. 623-630.
  9. Lee, S.-H., et al., Sagittal decompensation after corrective osteotomy for lumbar degenerative kyphosis: classification and risk factors. Spine, 2011. 36(8): p. E538-E544.
  10. Lee, C.-S., Y.-T. Kim, and E. Kim, Clinical study of lumbar degenerative kyphosis. Journal of Korean Society of Spine Surgery, 1997. 4(1): p. 27-35.

Comment 2) 

Discussion:

Please clearly describe why this research topic was chosen.

Please describe in detail why this study is so important and unique.

 Response 2 : Thank you for your valuable feedback. We have thoroughly reviewed your comments and have revised and enhanced the discussion and conclusion sections to incorporate your input.

The following sentences and references have been added for detailed explanation.

Page 11, Discussion, Line 292-305

The cortical bone plays a vital role in the load-bearing capacity of the vertebral body. As age-related bone mass loss progresses, the load-carrying capacity of the cortical bone increases significantly[34], rendering the cortex more susceptible to fractures. Fractures of the cortical bone within the vertebral body generally become more prevalent with aging, wherein the deformation of the cortical bone proves to be more sensitive to aging than the nucleus pulposus, trabecular matrix, bone endplates, and posterior elements[35, 36]. In our study, the stress on the cortical bone, annular fibrosus, and nucleus pulposus is reduced in postures with proper spinal alignment due to the adequate maintenance of lumbar lordosis. However, in postures like slumped sitting on a chair and sitting on the floor with reduced lumbar lordosis, the load on the cortical bone significantly increases compared to upright sitting and standing. If an individual continues to adopt a posture with reduced lumbar lordosis as they age, the deformation of the cortical bone will accelerate in addition to the deformation caused by aging. Consequently, the risk of cortical bone fractures within the vertebral body will escalate.

  1. Cao, K.D., M.J. Grimm, and K.-H. Yang, Load sharing within a human lumbar vertebral body using the finite element method. Spine, 2001. 26(12): p. e253-e260.
  2. Lu, Y., et al., Strain changes on the cortical shell of vertebral bodies due to spine ageing: A parametric study using a finite element model evaluated by strain measurements. Proceedings of the Institution of Mechanical Engineers, Part H: Journal of Engineering in Medicine, 2013. 227(12): p. 1265-1274.
  3. Christiansen, B.A., et al., Mechanical contributions of the cortical and trabecular compartments contribute to differences in age‐related changes in vertebral body strength in men and women assessed by QCT‐based finite element analysis. Journal of Bone and Mineral Research, 2011. 26(5): p. 974-983.

Page 12, Discussion, Line 322-342

Lumbar degenerative kyphosis (LDK) refers to a sagittal misalignment of the lumbar spine caused by unique lifestyle habits like prolonged squatting in agricultural work and engaging in daily activities on the floor[40]. Previous studies have indicated that LDK is primarily prevalent in rural regions of East Asia, such as Korea and Japan[19, 20, 40]. Patients diagnosed with LDK typically exhibit extensive degenerative alterations in the lower lumbar intervertebral discs and posterior joints spanning from the L2 to S1 levels, alongside atrophy and fatty changes in the lumbar extensors[20]. LDK has garnered considerable attention through numerous publications in Korea and Japan[19, 20, 40]. In contrast, LDK has been infrequently documented in Western countries; however, within Asian nations, LDK ranks among the most prevalent spinal deformities, likely due to distinct lifestyles and work-related postures.

The etiology of LDK is attributed to degenerative changes within the discs, encompassing disc space narrowing, vacuum phenomena in the discs, and thickening of the posterior joints, which collectively lead to a loss of lumbar lordosis[41]. Consistent with our study findings, the continued adoption of farming or daily life practices with reduced lumbar lordosis, such as slumped sitting in chairs or sitting on the floor, will escalate the load on the intervertebral disc and cortical bone. This increased load can result in disc degeneration, further disc space narrowing, and acceleration of vacuum phenomena within the disc. Moreover, in elderly women affected by osteoporosis, the load on the cortical bone is also heightened, consequently amplifying the risk of vertebral body fractures.

  1. Lee, S.-H., et al., Sagittal decompensation after corrective osteotomy for lumbar degenerative kyphosis: classification and risk factors. Spine, 2011. 36(8): p. E538-E544.
  2. Lee, C.-S., Y.-T. Kim, and E. Kim, Clinical study of lumbar degenerative kyphosis. Journal of Korean Society of Spine Surgery, 1997. 4(1): p. 27-35.
  3. Takemitsu, Y., et al., Lumbar degenerative kyphosis. Clinical, radiological and epidemiological studies. Spine, 1988. 13(11): p. 1317-1326.
  4. Jang, J.-S., et al., Lumbar degenerative kyphosis: radiologic analysis and classifications. Spine, 2007. 32(24): p. 2694-2699.

Page 11, Discussion, Line 276-291

< Sentence before improvement>

In this study, we confirmed that a large amount of von Mises stress is applied to the spinal structure in postures where lumbar lordosis is reduced, such as slumped sitting on a chair or sitting on the floor. This result is in agreement with previous studies. However, in previous studies, intradiscal pressure was measured by inserting a pressure sensor into pig vertebrae or directly into a human disc. These pioneering findings form the basis of our current knowledge of the in vivo loading conditions on the human spine but indicate that further studies are needed to better understand the mechanical loading of the lumbar spine. In this study, FEM analysis based on the physical properties of the human body was used.

We found that the stress values on the annulus fiber and nucleus pulposus were similar in both standing and erect sitting on a chair postures regardless of movement, likely because lumbar lordosis does not differ between these two postures, suggesting no significant difference in pressure on the spine and discs. Previous studies also showed conflicting results in the comparison of intravertebral disc pressure between an erect sitting posture on a chair and a standing posture, but no significant difference was found.

Slumped sitting on a chair and sitting on the floor showed that a large pressure difference was applied to the spine upon sitting compared to that with standing, in line with the findings of several studies. In addition, there was a marked pressure difference in slumped sitting when compared to that during erect sitting on a chair. Our results con-firmed the findings of previous studies, indicating that as lumbar lordosis decreased, the pressure on the intervertebral disc increased. This relationship was observed even during sitting, likely due to variations in waist posture. Our study provides evidence that maintaining an erect sitting posture can reduce the burden on the back, which is particularly important in modern society, where many people spend most of their day sitting and are at risk of back pain and related conditions.

< Sentence after improvement>

In this study, we found that the stress values of the annulus fibers and nucleus pulposus in both the erect sitting on a chair posture and the standing posture were similar regardless of the movement, which suggests that there is no significant difference in the stress values of the annulus fibers and nucleus pulposus because the lumbar lordosis is not different between the two postures. This is similar to results comparing intervertebral disc pressures in the erect sitting on a chair posture and the standing posture in static conditions, where no significant differences were found[12, 24, 31]. We found no significant difference in the stress values of the annulus fibers and nucleus pulposus in the standing and erect sitting on a chair postures with or without movement. However, the stress values of the annulus fibers and nucleus pulposus in the slumped sitting on a chair and sitting on the floor postures was significantly different when compared to the standing posture. These results were similar to previous studies conducted in each posture in a static state[12, 31-33]. Ultimately, we found that the stress of the annulus fibers and nucleus pulposus increased in the posture with reduced lumbar lordosis even in a static state, and that the stress increased further in the posture with reduced lumbar lordosis when moving in each posture.

  1. Nachemson, A., The load on lumbar disks in different positions of the body. Clinical Orthopaedics and Related Research (1976-2007), 1966. 45: p. 107-122.
  2. Li, J.-Q., et al., Comparison of in vivo intradiscal pressure between sitting and standing in human lumbar spine: a systematic review and meta-analysis. Life, 2022. 12(3): p. 457.
  3. Wilke, H.J., et al., New in vivo measurements of pressures in the intervertebral disc in daily life. Spine (Phila Pa 1976), 1999. 24(8): p. 755-62.
  4. Andersson, G., R. Ortengren, and A. Nachemson, Quantitative studies of the load on the back in different working-postures. Scandinavian journal of rehabilitation medicine. Supplement, 1978. 6: p. 173-181.
  5. Lord, M.J., et al., Lumbar lordosis. Effects of sitting and standing. Spine (Phila Pa 1976), 1997. 22(21): p. 2571-4.

Page 12, Discussion, Line 343-358

< Sentence before improvement>

A benefit of lumbar lordosis was suggested by a previous study, which noted an as-sociation between reduced lumbar lordosis and increased disc degeneration at S-L5 [32]. Another study also reported an inverse relationship between lordosis and intradiscal pressure [33]. Although Nachemson’s study did not specifically identify lordosis as a de-terminant of intravertebral pressure, a careful review showed that intravertebral pressure increased at locations where lordosis was reduced [26]. A study of the relationship be-tween lumbar lordosis and sitting found that the most important factor for back pain caused by long-term sitting was a reduction in the trunk-thigh angle resulting from the flattening of the lumbar spine curve [34].

For those who spend most of their day sitting, our results indicate that proper sitting postures should be considered. The results of our study confirmed that the pressure ap-plied to the lumbar spine was reduced in the erect sitting posture while maintaining lumbar lordosis. In other studies, the load applied to the lumbar spine was lower when sitting with support than when sitting without support because the weight of the upper body is partially transferred to the chair back and armrests, reducing the load on the spine [27,35,36].

< Sentence after improvement>

A previous study suggested a benefit of lumbar lordosis, noting an association between reduced lumbar lordosis and increased disc degeneration at S-L5[42]. Similarly, Another study reported an inverse correlation between lordosis and intradiscal pressure [43]. Although Nachemson’s study did not explicitly establish lordosis as a determinant of intravertebral pressure, a careful examination revealed that intravertebral pressure increased at locations where lordosis was diminished [12]. An investigation into the relationship between lumbar lordosis and sitting highlighted that the most important factor for back pain caused by long-term sitting was the reduction in the trunk-thigh angle resulting from the flattening of the lumbar spine curve [44].

For individuals who spend a majority of their day sitting, our findings emphasize the importance of adopting proper sitting postures. The results of our study confirmed that the stress applied to the lumbar spine decreased in an erect sitting posture that maintains lumbar lordosis. Other studies have also indicated that the load on the lumbar spine is lower when sitting with support compared to sitting without support. This is due to the upper body's weight being partially transferred to the chair's back and armrests, consequently alleviating the load on the spine [16, 45, 46].

  1. Nachemson, A., The load on lumbar disks in different positions of the body. Clinical Orthopaedics and Related Research (1976-2007), 1966. 45: p. 107-122.
  2. Wilke, H.J., et al., New in vivo measurements of pressures in the intervertebral disc in daily life. Spine, 1999. 24(8): p. 755-762.
  3. Farfan, H., R. Huberdeau, and H. Dubow, Lumbar intervertebral disc degeneration: the influence of geometrical features on the pattern of disc degeneration—a post mortem study. JBJS, 1972. 54(3): p. 492-510.
  4. Adams, M. and W. Hutton, The effect of posture on the lumbar spine. The Journal of bone and joint surgery. British volume, 1985. 67(4): p. 625-629.
  5. Williams, M.M., et al., A comparison of the effects of two sitting postures on back and referred pain. Spine, 1991. 16(10): p. 1185-1191.
  6. Aota, Y., et al., Effectiveness of a lumbar support continuous passive motion device in the prevention of low back pain during prolonged sitting. Spine (Phila Pa 1976), 2007. 32(23): p. E674-7.
  7. Moriguchi, C.S., T.O. Sato, and H. Coury, An Instrumented Workstation to Evaluate Weight-Bearing Distribution in the Sitting Posture. Saf Health Work, 2019. 10(3): p. 314-320.

Page 13, Conclusion, Line 385-393

< Sentence before improvement>

In conclusion, this study highlights that maintaining an erect sitting posture on a chair can effectively reduce lumbar spine pressure to a level comparable to standing for individuals who spend a significant amount of time sitting. Moreover, it is crucial to avoid sitting on the floor, a common practice in Asian culture, as it substantially increases intra-discal pressure. Instead, adopting an erect sitting position on a chair and making lifestyle changes to promote proper lumbar lordosis maintenance will play a pivotal role in preventing various degenerative disc diseases.

< Sentence after improvement>

In conclusion, this study emphasizes that maintaining an erect sitting posture on a chair can reduce von Mises stress on the annulus fibrosus, nucleus pulposus, and cortical bone to a level similar to that of standing, especially important for modern individuals who spend a significant amount of time sitting. Additionally, the posture of sitting on the floor, common in East Asian cultures, leads to increased von Mises stresses in the annulus fibrosus, nucleus pulposus, and cortical bone when compared to the upright sitting posture on a chair. This posture seems to be associated with the development of lumbar degenerative kyphosis, frequently observed in East Asia. Adopting a lifestyle that involves sitting upright in a chair and maintaining proper lumbar lordosis will play a pivotal role in preventing various degenerative disc diseases and spinal deformities.

Round 2

Reviewer 1 Report

The authors have made many modifications to the paper, which has improved its quality.

However, I noticed that there are still some shortcomings.

In the boundary and loading conditions, flexibility, lateral bonding, and axial rotation movements of the lumbar spine were introduced, but the influence of lumbar muscle force was not considered, which had a significant impact on the results (https://doi.org/10.3390/bioengineering10010067). Although you mentioned this deficiency at the end of Discussion section, it is not enough and it is still necessary to discuss the impact of this simplification on the conclusion.

The result section lacks a stress cloud map, which cannot reflect the location of stress concentration.

The same result only needs to be expressed in one form from the graph or table. Repetitive expression is redundant.

Some references are too outdated, and more should be introduced to the latest literature.

Author Response

Dear Reviewer #1

Thank you very much for your kind letter and comments concerning our manuscript entitled

“Biomechanical Effects of Different Sitting Postures and Physiologic Movements on the Lumbar Spine: A Finite Element Study”

The comments were valuable and very helpful in critically revising and improving our paper. We have discussed the comments carefully and have made corresponding corrections which we hope will be met with approval. The following are responses to your comments

Comment 1)

In the boundary and loading conditions, flexibility, lateral bonding, and axial rotation movements of the lumbar spine were introduced, but the influence of lumbar muscle force was not considered, which had a significant impact on the results (https://doi.org/10.3390/bioengineering10010067). Although you mentioned this deficiency at the end of Discussion section, it is not enough and it is still necessary to discuss the impact of this simplification on the conclusion.

Response 1 :

Thank you for your thoughtful feedback regarding the importance of lower back muscle strength. We greatly value your keen observations and insightful suggestions.

You are absolutely correct in pointing out that lumbar muscle strength can significantly impact biomechanical outcomes, and we acknowledge this limitation in our study. We recognize the need for a more comprehensive discussion on how these simplifications affect our conclusions.

In response to your feedback, we have made revisions to the Discussion section to provide a more detailed analysis of how excluding the lower trunk muscles from our modeling has influenced our findings.

Including the multifidus muscle in Finite Element (FE) modeling has a significant effect on reducing the error associated with overestimating compression loading on the lumbar spine in muscle-free simulations[1]. Research has demonstrated that muscles responsible for stabilizing the lumbar spine, such as the multifidus muscle, play a crucial role in mitigating shear stress on the intervertebral discs[1]. These muscles achieve this by stabilizing the spine during lumbar movements. Consequently, when these lumbar spine stabilizing muscles are more active, the stress placed on the lumbar spine is reduced.

Previous studies[2,3] have investigated muscle activation patterns responsible for spinal stabilization in various sitting postures. These studies have uncovered that core muscles, which contribute significantly to spinal stability, are less activated in a slumped sitting posture compared to an erect sitting posture. This finding suggests that in postures with reduced lumbar lordosis, such as slumped sitting in a chair, there is a decrease in the activation of spinal stabilizing muscles. Consequently, this diminished activation leads to increased stress on lumbar structures, including the cortical bone, annulus fibers, and nucleus pulposus.

Conversely, in an erect sitting posture on a chair, the activation of core muscles responsible for stabilizing the spine increases. This increased activation serves to further reduce the stresses placed on the cortical bone, annulus fibers, and nucleus pulposus.

In summary, accounting for muscle activation in Finite Element (FEM) studies is expected to reveal a more significant difference in stresses on these lumbar structures between erect sitting and slumped sitting on a chair than what our current experimental results indicate.

  1. Wang, K., et al., The role of multifidus in the biomechanics of lumbar spine: A musculoskeletal modeling study. Bioengineering, 2023. 10(1): p. 67.
  2. Waongenngarm, P., B.S. Rajaratnam, and P. Janwantanakul, Perceived body discomfort and trunk muscle activity in three prolonged sitting postures. Journal of physical therapy science, 2015. 27(7): p. 2183-2187.
  3. Wong, A.Y., et al., Do different sitting postures affect spinal biomechanics of asymptomatic individuals? Gait & posture, 2019. 67: p. 230-235.

The following sentences and references have been added for detailed explanation.

Page 15, Discussion, Line 443-461

The inclusion of the multifidus muscle in FEM modeling can reduce the error of overestimating the compression loading of the lumbar spine in muscle-free simulations. It has been found that muscles which stabilize the lumbar spine, such as the multifidus muscle, can decrease the shear forces acting on the intervertebral discs by stabilizing the spine during lumbar spine movements[53]. Consequently, it can be predicted that the more active the lumbar spine stabilizing muscles are, the less load is placed on the lumbar spine. In addition, previous studies have examined muscle activation patterns responsible for spinal stabilization in different sitting postures and found that core muscles contributing to spinal stability were less activated in slumped sitting compared to erect sitting[54, 55]. Applying these findings to our results, we interpret that in postures with reduced lumbar lordosis, such as slumped sitting in a chair, there is reduced activation of spinal stabilizing muscles and consequently increased stress on lumbar structures, including cortical bone, annulus fibers, and nucleus pulposus. Consequently, in the erect sitting posture on a chair, the activation of the core muscles that stabilize the spine increases, further reducing the stresses on the cortical bone, annulus fibers, and nucleus pulposus. Therefore, if an FEM study could be performed taking into account the activation of the muscles, we would expect to observe a greater difference in the stresses on the cortical bone, annulus fibers, and nucleus pulposus between erect sitting on a chair and slumped sitting on a chair than our experimental results indicate

  1. Wang, K., et al., The role of multifidus in the biomechanics of lumbar spine: A musculoskeletal modeling study. Bioengineering, 2023. 10(1): p. 67.
  2. Waongenngarm, P., B.S. Rajaratnam, and P. Janwantanakul, Perceived body discomfort and trunk muscle activity in three prolonged sitting postures. Journal of physical therapy science, 2015. 27(7): p. 2183-2187.
  3. Wong, A.Y., et al., Do different sitting postures affect spinal biomechanics of asymptomatic individuals? Gait & posture, 2019. 67: p. 230-235.

Comment 2)

The result section lacks a stress cloud map, which cannot reflect the location of stress concentration.

Response 2 :

Thank you for your valuable feedback. We appreciate your insight into the limitations of our results section. Your observation regarding the absence of a stress cloud map is well-taken. We inserted stress cloud maps of the cortical bone, annulus fiber, and nucleus pulposus for each posture during flexion, lateral bending, and axial rotation motions as Figure 5, Figure 6, and Figure 7, respectively.

We genuinely appreciate your constructive feedback, which will undoubtedly improve the robustness of our study.

Thank you once again for your time and thoughtful review.

The following figures have been added for detailed explanation.

Figure. 5. The von Mises stress distribution in the cortical bone, annulus fiber, and nucleus pulposus of the lumbar spine (L1-L5) during four postures (Standing, Erect sitting on a chair, Slumped sitting on a chair, Sitting on the floor) during flexion motion is displayed. (Unit: MPa)

Figure. 6. The von Mises stress distribution in the cortical bone, annulus fiber, and nucleus pulposus of the lumbar spine (L1-L5) during four postures (Standing, Erect sitting on a chair, Slumped sitting on a chair, Sitting on the floor) during lateral bending motion is displayed. (Unit: MPa)

Figure. 7. The von Mises stress distribution in the cortical bone, annulus fiber, and nucleus pulposus of the lumbar spine (L1-L5) during four postures (Standing, Erect sitting on a chair, Slumped sitting on a chair, Sitting on the floor) during axial rotation motion is displayed. (Unit: MPa)

Comment 3)

The same result only needs to be expressed in one form from the graph or table. Repetitive expression is redundant.

Response 3 :

Thank you for your feedback and for pointing out the redundancy in our presentation of results. We appreciate your keen observation. We agree that repetitive expressions can be redundant and may affect the clarity and conciseness of our paper.

We confirmed that Tables 2, 3, and 4 in our previous manuscript and Figure 5 displayed identical results. As per your comment, Figure 5 in our previous manuscript has been removed.

Thank you once again for your time and thoughtful review.

Comment 4)

Some references are too outdated, and more should be introduced to the latest literature.

Response 4 :

Thank you for your thoughtful feedback on the references in the manuscript. Your input is greatly appreciated, and we recognize how important it is to base our research on the latest literature.

We have certainly reviewed the references and endeavored to introduce more recent and relevant studies into the manuscript. We have updated some outdated references with more recent ones, and we have also added new references, which should help strengthen the overall quality and relevance of the study.

Thank you again for taking the time to provide us with your valuable suggestions.

Here's the list of references we added

  1. Waxenbaum, J.A., et al., Anatomy, back, lumbar vertebrae. 2017.
  2. Galbusera, F. and H.-J. Wilke, Biomechanics of the spine: Basic concepts, spinal disorders and treatments. 2018: Academic Press.
  3. Sparrey, C.J., et al., Etiology of lumbar lordosis and its pathophysiology: a review of the evolution of lumbar lordosis, and the mechanics and biology of lumbar degeneration. Neurosurgical focus, 2014. 36(5): p. E1.
  4. Li, J.Q., et al., Comparison of In Vivo Intradiscal Pressure between Sitting and Standing in Human Lumbar Spine: A Systematic Review and Meta-Analysis. Life (Basel), 2022. 12(3).
  5. Palepu, V., S.D. Rayaprolu, and S. Nagaraja, Differences in trabecular bone, cortical shell, and endplate microstructure across the lumbar spine. International journal of spine surgery, 2019. 13(4): p. 361-370.
  6. Miura, T., et al., Relationship between intervertebral disc compression force and sagittal spinopelvic lower limb alignment in elderly women in standing position with patient-specific whole body musculoskeletal model. International Journal of Environmental Research and Public Health, 2022. 19(24): p. 16452.
  7. Pourabbas Tahvildari, B., et al., The impact of spino-pelvic parameters on pathogenesis of lumbar disc herniation. Musculoskelet Surg, 2022. 106(2): p. 195-199.
  8. Mosabbir, A., Mechanisms behind the development of chronic low back pain and its neurodegenerative features. Life, 2022. 13(1): p. 84.
  9. Waongenngarm, P., B.S. Rajaratnam, and P. Janwantanakul, Perceived body discomfort and trunk muscle activity in three prolonged sitting postures. Journal of physical therapy science, 2015. 27(7): p. 2183-2187.
  10. Wong, A.Y., et al., Do different sitting postures affect spinal biomechanics of asymptomatic individuals? Gait & posture, 2019. 67: p. 230-235.
